# Enhancing Starch−Based Packaging Materials: Optimization of Plasticizers and Process Parameters

**DOI:** 10.3390/ma16175953

**Published:** 2023-08-30

**Authors:** Yue Wu, Rongji Tang, Anfu Guo, Xiaodong Tao, Yingbin Hu, Xianliang Sheng, Peng Qu, Shaoqing Wang, Jianfeng Li, Fangyi Li

**Affiliations:** 1School of Mechanical and Automotive Engineering, Liaocheng University, Liaocheng 252000, China; 2020207515@stu.lcu.edu.cn (Y.W.); 2120230119@stu.lcu.edu.cn (R.T.); taoxiaodong2023@163.com (X.T.); 2220230101@stu.lcu.edu.cn (X.S.); qupeng@lcu.edu.cn (P.Q.); wangshaoqing@lcu.edu.cn (S.W.); 2Department of Mechanical and Manufacturing Engineering, Miami University, Oxford, OH 45056, USA; huy70@miamioh.edu; 3School of Mechanical Engineering, Shandong University, Jinan 250061, China; ljf@sdu.edu.cn (J.L.); lifangyi@sdu.edu.cn (F.L.)

**Keywords:** plasticizer optimization, thermoplastic starch, Box–Behnken design, response surface method, starch−based composite packaging material

## Abstract

In order to actively promote green production and address these concerns, there is an urgent need for new packaging materials to replace traditional plastic products. Starch−based packaging materials, composed of starch, fiber, and plasticizers, offer a degradable and environmentally friendly alternative. However, there are challenges related to the high crystallinity and poor compatibility between thermoplastic starch and fibers, resulting in decreased mechanical properties. To address these challenges, a novel approach combining plasticizer optimization and response surface method (RSM) optimization has been proposed to enhance the mechanical properties of starch−based packaging materials. This method leverages the advantages of composite plasticizers and process parameters. Scanning electron microscopy and X-ray crystallography results demonstrate that the composite plasticizer effectively disrupts the hydrogen bonding and granule morphology of starch, leading to a significant reduction in crystallinity. Fourier transform infrared spectroscopy results show that an addition of glycerol and D−fructose to the starch can form new hydrogen bonds between them, resulting in an enhanced plasticizing effect. The optimal process parameters are determined using the RSM, resulting in a forming temperature of 198 °C, a forming time of 5.4 min, and an AC content of 0.84 g. Compared with the non−optimized values, the tensile strength increases by 12.2% and the rebound rate increases by 8.1%.

## 1. Introduction

Packaging materials derived from petroleum−based plastics are non−biodegradable, leading to severe pollution of the environment upon which humans depend [1,2,3] and posing a significant threat to human health [4,5,6]. As a result, numerous researchers have turned their attention to biomass materials, which are obtained from natural sources and can undergo complete degradation. Examples of such biomass materials include corn, wheat, sorghum, and others [7,8,9]. By combining these materials with fibers and incorporating a foaming agent, it is possible to prepare a biomass packaging material with a cellular structure [10].

Among various biomass materials, starch has gained recognition as an environmentally friendly material due to its easy availability, economic benefits, and ability to completely decompose into carbon dioxide and water under natural conditions [11,12]. Starch molecular chains contain a significant number of hydroxyl (–OH) groups, which readily undergo hydrogen bonding with each other, promoting the formation of a dispersed microcrystalline structure known as the double helix structure [13]. During the gelatinization process of starch, water molecules disrupt the hydrogen bonding mechanism of starch, causing the double helix structure to unwind and stretching the starch molecular chains [14]. However, at this stage, the starch molecular chains are not stable, and over time, the hydrogen bonding tends to reform, leading to the formation of different ordered structures, known as retrogradation [15]. The presence of glucoside bonds (C−O−C) and –OH groups on the starch molecular chains results in strong intermolecular forces, low molecular chain mobility, and limited relative sliding. This unique structure gives starch a decomposition temperature lower than its melting temperature, making it unsuitable for direct thermoplastic processing [16,17]. Therefore, the structure of starch needs to be modified or disrupted.

Currently, the most common method of disrupting the hydrogen bonding mechanism and modifying the structure of starch is through the use of plasticizers, which enhance the thermoplastic processing performance and ensure the compatibility of starch with other polymers [18,19,20]. Avila−Martin et al. [19] extracted starch film from canna roots and introduced citric acid and whey protein isolate as modifiers. The results demonstrated that both citric acid and whey protein isolate effectively plasticized the starch film, leading to more uniform hydrogen bonding and improved material durability. Zeng et al. [20] investigated the modification of natural corn starch (NCS) using glycerol (GLY) and urea as composite plasticizers. The study revealed that the addition of 15 parts per hundred resulted in plasticized NCS and a composite with favorable mechanical properties. Commonly used plasticizers can be divided into polyols and amines. Polyols include water, glycerol, ethylene glycol, sorbitol, etc., while amines include urea, formamide, and so on. Among various plasticizers, GLY, as a safe and harmless small−molecule polyol, exhibits high mobility and can disrupt the original hydrogen bonding mechanism of starch, reducing its crystallinity and glass transition temperature, thus conferring thermoplastic processing properties [21,22]. However, the hydrogen bond between GLY and starch is relatively weak, making it prone to retrogradation after long−term storage [23]. D−fructose, as a monosaccharide plasticizer with a larger molecular weight, can form better bonds with hydrogen in starch [24]. Furthermore, Saberi et al. [25] suggested that sugar plasticizer molecules interact with water molecules during the plasticization process, inhibiting the recrystallization process and preventing retrogradation. Therefore, in this investigation, GLY and D−fructose are selected as plasticizers to investigate the effects of individual and combined effects on starch.

The investigation of plasticizers has facilitated the development of thermoplastic starch (TPS). However, in the actual preparation and production of starch−based packaging materials, it is necessary to incorporate natural fibers into the starch matrix. Nevertheless, the high crystallinity and poor biocompatibility between the two components significantly impact their mechanical properties [26,27,28]. In order to achieve improved mechanical performance, numerous scholars have carried out a series of studies. Zhang et al. [29] prepared oxidized starch using an oxidant and successfully utilized GLY as a plasticizer to create thermoplastic oxidized starch with favorable rheology and toughness. The study revealed the formation of new hydrogen bonds between the plasticizer, oxidant, and starch, resulting in a combination of oxidation and plasticization effects. This led to changes in the crystal structure of oxidized starch and significantly improved its pressure resistance. Chen et al. [30] produced a starch composite through oxidative esterification. In comparison to natural starch composites, the starch composite produced through oxidative esterification exhibited higher structural damage, lower crystallinity, and notable improvements in toughness and tensile strength. Zha et al. [31] grafted hydrophilic polyacrylic acid branches onto a starch acetate framework using water as the medium. The study demonstrated that the hydrophilic polyacrylic acid branches effectively reduced the crystallinity of starch acetate, mitigated film brittleness, and enhanced the adhesion of starch acetate to fibers and films. Peidayesh et al. [32] employed a dual crosslinking strategy by in−situ reacting starch, glycerol, and epichlorohydrin to generate both non−covalent and covalent bonds. The tensile strength of the TPS obtained through this method showed a significant improvement compared to conventional TPS.

The previous research demonstrates that the predominant focus has been on optimizing plasticizers. However, there have been limited studies concerning the amalgamation of plasticizer optimization and the optimization of process parameters for thermal cavity forming. Ararsa et al. [33] employed response surface analysis to optimize the proportions of starch, glycerin, and acetic acid, yet they did not address the optimization of process parameters during fabrication. In another instance, Perez−Chavez et al. [34] employed an extrusion process to manufacture flame−retardant polymer composite materials. They utilized the response surface methodology to optimize factors such as keratin fiber length, blending temperature, and rotation speed during the molding process, aiming to strike a balance between flame retardancy and mechanical properties. Nevertheless, this approach pertained solely to material optimization. It is important to note that molding equipment constitutes a crucial component in material processing. Thus, optimization efforts should encompass not only the selection and proportions of plasticizers but also the refinement of molding equipment’s process parameters.

In this current study, a blend of glycerin and D−fructose was incorporated into corn starch to create a novel thermoplastic starch (TPS). The intention was to disrupt the crystal structure of TPS and diminish the crystallization peak, thereby augmenting the plasticizing effect. Subsequently, the formulated TPS was combined with pre−fabricated sisal fibers and processed using a thermal cavity forming machine. The investigation delved into the impacts of forming temperature, forming time, and foaming agent concentration on tensile strength and resilience during the thermal cavity forming process. Through response surface analysis, optimal process parameters were determined. Consequently, the objective of this study was to produce an innovative composite material employing a starch matrix, wherein thermoplastic corn starch serves as the matrix and sisal fibers function as reinforcing agents. Furthermore, the study aimed to fine−tune the processing parameters using response surface analysis.

## 2. Materials and Experimental Procedures

### 2.1. Materials Preparation and Experimental Set−Up

In this study, NCS (Huachen Starch Sugar Co., Ltd., Shijiazhuang, Hebei, China) was used as the substrate. Sisal fibers (Li Qiang Fiber Industry Co., Ltd., Chongzuo, Guangxi, China) with an average length of 5 mm and an average diameter of 60 μm were used as reinforcement material. GLY (C3H8O3, Fuyu Fine Chemical Co., Ltd., Tianjin, China) and D−fructose (Chengchuan Technology Co., Ltd., Jinan, Shandong, China) were used as the plastic agents. The molecular structures of GLY and D−fructose are shown in Figure 1. AR−grade azodiacarbonamide (AC, Ruian in rubber and plasticizing additives Co., Ltd., Ruian, Zhejiang, China) was used as the blowing agent. AR−grade tricarboxylic acid was used as the release agent (Guangcheng Chemical Co., Ltd., Tianjin, China).

The preparation process for the starch−based packing structure is provided in Figure 2. Firstly, the NCS and distilled water (Shuangshuang Chemical Co., Ltd., Yantai, Shandong, China) were mixed with a mass ratio of 1: 3 in a beaker. Then, a glass rod was used to stir the mixture slowly until the starch was completely gelatinized. Then, the mixture was kept at 86 °C in a water bath (Joanlab, Huzhou, Zhejiang, China) and stirred for 30 min using a precision force mixer (JJ−1, Shibo Experimental Instrument Factory, Changzhou, Jiangsu, China). Afterward, the plasticizer (GLY, D−fructose) was added to the beaker at different mass ratios according to Table 1. The new mixture was stirred at high speed until the starch was completely gelatinized. Then, the new mixture was kept at 86 °C in a water bath for 12 h.

It is worth noting that the moisture content and relative humidity may adversely affect the gelatinization and crystallization of the starch. To avoid these effects, a dehumidifier was used to keep the indoor humidity below 40%. The resulting gelatinized starch was dried in a vacuum drying oven at 80 °C for 24 h to obtain TPS, as shown in Figure 2a. Then, the sisal fiber was soaked in a 2 wt.% NaOH solution for 8 h, removed, washed repeatedly with distilled water until the pH value reached 7, and dried in a drying box at 86 °C for 12 h to obtain the alkalized fiber, as shown in Figure 2b. Finally, the TPS, alkalized fiber, and blowing agent were thoroughly mixed in a high−speed mixer with a mass ratio of 10:3:0.1 to obtain a composite slurry, as shown in Figure 2c. The composite paste was placed in the lower mold of a self−made double−column simplex thermal cavity forming machine (Figure 3), and the mold temperature was set at 198 °C. Subsequently, the upper mold was moved down, the mold was closed, and the slurry was deformed, as depicted in Figure 3. The resulting starch−based packaging material is shown in Figure 2d.

The foaming process is depicted in Figure 4. In the transition from the first stage to the second stage, the viscosity of TPS gradually decreases as the temperature increases. In the second stage, AC decomposes gradually, releasing water vapor, nitrogen, and carbon monoxide. These gases facilitate pore growth. In the third stage, TPS is gradually compressed and tightly wrapped around the sisal fibers during the solidification process. Eventually, a cellular structure is formed with sisal fibers as the framework and starch as the matrix.

### 2.2. Material Characterizations

A portion of the TPS prepared in 2.1 was subjected to drying in a drying box at 60 °C for 24 h to remove moisture. After thorough grinding, it was strained through a 200−mesh screen and left at room temperature for 30 days. A sample of 1 mg of TPS was taken and mixed with 150 mg of potassium bromide, then pressed into a transparent sheet. Fourier Transform Infrared Spectroscopy (FT−IR) (NICOLET−IR200, Thermo Fisher Scientific Company, Shanghai, China) was employed to detect and analyze the effect of plasticizers on the hydrogen bonds of starch within the spectral range of 500–4000 cm^−1^. Scanning was recorded with a resolution of 2 cm^−1^. X-ray diffraction (XRD) (SmartLab 9 kW, Tokyo, Japan) was utilized to analyze the changes in crystal types before and after starch modification. The scanning speed was set at 5°/min, and the analysis angle ranged from 10° to 80°. Scanning Electron Microscopy (SEM) (Phenom XL−SE, Eindhoven, The Netherlands) was employed to examine the impact of plasticizers on the micromorphology of starch.

### 2.3. Mechanical Performance Testing

Prior to tensile and compression tests, the samples were dried in an oven at 86 °C for 24 h to remove moisture. Tensile specimens were prepared following the guidelines outlined in GB/T9641−88. The tensile test was conducted using the Electronic Universal Material Experimental machine (KX WDW3200, Kexin Experimental Instrument Co., Ltd., Changchun, Jilin, China) at a constant speed of 5 mm/min. To minimize experimental errors, five trials were performed for each composite material. The compressed sample was prepared in accordance with the GB/T 8186−2008 standard, with a sample size of 25 mm (length) × 25 mm (width) × 5 mm (height). The compression test machine was set to a speed of 12 mm/min. The original thickness of the sample was maintained, and the load was gradually applied in the direction perpendicular to the thickness. After the sample underwent a 30% deformation, a stable load was sustained for a duration of 3 min. Afterward, the sample was unloaded and left for 30 s. The thickness of the compressed specimen was measured using a spiral micrometer, and the rebound rate was calculated. Each group of experiments was repeated three times, with an interval of 1 min between each experiment. The average result from the three repetitions was considered the rebound rate for that particular sample group. The formula for calculating the rebound rate is as follows [35]:(1)tj=Tj−Tj2Ti2×100%
where, $t_{j}$ is the rebound rate after j experiments; $T_{j}$ is the thickness of the specimen after j times compression, mm; and $T_{i}$ is the original thickness of the sample, mm. i and j are the number of experiments.

### 2.4. Process Parameter Optimization

A starch−based biomass packaging material with a bubble cell structure was prepared using TPS as the matrix, alkalized sisal fiber as the reinforcement, and azodicarbonamide (AC) as the blowing agent. The thermal cavity−forming process was employed for fabrication. The RSM was utilized to optimize the process parameters, with the variables being forming temperature, forming time, and AC content. The response values considered for optimization were rebound rate and tensile strength. A regression model was fitted, and variance analysis was conducted to optimize the forming process. Each factor was designed with three levels, as presented in Table 2. The experiment was performed 17 times, with the central point repeated 5 times.

The three input variables were optimized based on the following steps.

(1)Establishing a general mathematical relationship between the input variables and response, which can be expressed as follows:
(2)$$Y=f\left(X_{1},\;X_{2},\;X_{3}\;\ldots \;\ldots ,\;X_{n}\right)$$
where Y is the response, f represents the mathematical relationship, and $X_{1}$, $X_{2}$, $X_{3}$… …, $X_{n}$ are input variables.(2)Looking for mathematical models. Based on mathematical analysis, the existing McLaughlin or Taylor expansions can meet convergence conditions. According to Samir Charola et al. [36], the experimental data can be fitted into a second−order polynomial model, namely:(3)$$Y=\beta_{0}+\sum_{i=1}^{k}\beta_{i}x_{i}\;+\sum_{i=1}^{k}\beta_{ii}x_{i}{}^{2}+\sum_{i_{i>j}}^{k}\sum_{j}^{k}\beta_{ij}x_{i}x_{j}$$
where $\beta_{0}$, $\beta_{i}$, $\beta_{ij}$, and $\beta_{ii}$ are the regression coefficients of the intercept, the linearity, the interaction, and the quadratic terms, respectively. $x_{i}$ and $x_{j}$ denote the input variables, and k represents the number of input variables selected for process optimization.(3)Using analysis of variance (ANOVA) to evaluate the significance of the prediction model. The reliability of the model can be determined by loss of fit (LOF) and *p* value. The actual values are the tensile strength and rebound rate measured by experiments. Based on the second−order polynomial model, the response surface and the contour plot, which describe the influences of two variables on a single target, can be obtained. The third input variable remains unchanged at the center point. Then, the effects of input variables on the response will be analyzed to obtain the highest rebound rate and tensile strength simultaneously.

## 3. Results and Discussion

### 3.1. The Plasticization Process of Starch

Starch is considered a natural polycrystalline system. Crystalline, submicron crystalline, and amorphous structures alternate to form the granular structure of starch. Under the combined action of plasticizers, temperature, and moisture, the inherent crystalline structure of starch is disrupted, resulting in a lower glass transition temperature and obtaining thermoplastic processing properties. The plasticization process of starch can be divided into four stages, as shown in Figure 5.

Stage I: Starch absorbs water and swells. Natural starch aggregates in the form of granules. When subjected to distilled water, the starch granules absorb water and swell, which is also known as reversible water absorption and swelling. During this process, water absorption occurs in the amorphous regions of the starch, while the crystalline regions remain intact. Once the water disperses, the starch can revert back to its original granular structure.

Stage II: Starch granules break. As the ambient temperature increases, under the combined action of plasticizers, mechanical agitation, and moisture, the starch granules begin to break.

Stage III: A new hydrogen bonding mechanism forms. Under the action of thermal and continuous mechanical agitation, plasticizer molecules enter the interior of starch granules, disrupting the existing hydrogen bonds. The entire molecules undergo relaxation, resulting in a destruction of the crystalline structure and an increase in the number of free branching chains, thereby increasing the free volume within the starch. At the same time, the plasticizer molecules provide a large number of –OH groups, making them more prone to interact with the –OH groups in starch. The resulting starch−plasticizer hydrogen bonding mechanism replaces the original starch−starch hydrogen bonding mechanism.

Stage IV: Plasticized starch forms. Under the influence of the new hydrogen bonding mechanism, starch molecules undergo rearrangement, leading to increased molecular chain mobility and a decrease in melting temperature. As a result, starch acquires processability.

In the plasticization process, plasticizers play a crucial role by primarily disrupting the hydrogen bonding in starch. The mechanism of their action is illustrated in Figure 6. Plasticizers enter the interior of starch and interact with the –OH groups on starch molecules, forming new hydrogen bonding mechanisms.

### 3.2. Effects of Plasticizer on TPS

#### 3.2.1. Effects of Plasticizer on Hydrogen Bond

Figure 7a shows the infrared spectral images of the NCS and the starch with different proportions of GLY plasticizer additions. In infrared spectrum detection, when the hydrogen bond is strengthened, the vibration frequency of the –OH group will decrease, and the vibration spectrum band will become wider. This can be attributed to the fact that the strength of the hydrogen bond can influence the vibration mode of the –OH group, thereby rendering it more resistant to excitation. The strength of the hydrogen bond is determined by the combined influence of the hydrogen bond length and angle, as well as the electric charge density at the center of the hydrogen bond. In addition, the establishment of a hydrogen bond elongates the bond length of the –OH group, increases the vibration absorption energy, and drives the absorbance shift to the low frequency. It is known that the frequency equals the energy of the wave multiplied by the speed of light. The more the number of waves moves, the stronger the hydrogen bond will be [37].

The infrared spectrum of NCS (black dashed line in Figure 7a) shows that the absorption peak of the −OH occurs at 3446 cm^−1^. With the increase in GLY content, the frequency of the absorption peak gradually decreases to 3441.1 cm^−1^, 3416.5 cm^−1^, 3404.6 cm^−1^, and 3423.7 cm^−1^, indicating that a new hydrogen bond is formed between GLY and NCS molecules. When the mass ratio between NCS and GLY is 10:3, the vibration frequency of the –OH group is the lowest, and the spectrum is wide, indicating that the hydrogen bond is the strongest and the plasticization effect is the best at this mass ratio. When the GLY content increases until the mass ratio between NCS and GLY reaches 10:4, the absorption peak of the –OH wave tends to shift to a higher frequency, indicating that an excessive GLY addition was not conducive to the plasticization process of NCS. This might be attributed to the fact that the excess GLY itself also forms hydrogen bonds, of which one –OH and the other –OH form a hydrogen bond, thus enhancing the stability of the GLY and reducing the plasticization effect.

In the plasticization process, there are a large number of –OH groups in starch molecules. In the process of forming covalent molecules between the hydrogen atoms in the –OH groups and the electronegative oxygen atoms, the electrons from the hydrogen atoms are shared on chemical bonds and are strongly attracted by neighboring atoms. Therefore, the electron clouds of the hydrogen atoms are away from the hydrogen nuclei and the shared electron pairs between atoms, and the hydrogen atoms show positive H^+^. GLY contains a significant number of electronegative oxygen atoms. In the process of penetrating starch granules, small molecular weights of GLY and H^+^ react to generate a magnetic field, which destroys the hydrogen bonds within starch. Due to the action of the van der Waals force [38], the GLY and H^+^ are directionally aligned with the formation of a new hydrogen bond structure.

Figure 7b shows the effects of D−fructose plasticizer contents on the infrared spectra of TPS. Results show that when the mass ratios between NCS and D−fructose plasticizer are 10:1, 10:2, 10:3, and 10:4, the absorption peak frequencies are located at 3423.8 cm^−1^, 3406.9 cm^−1^, 3422.5 cm^−1^, and 3432.9 cm^−1^, respectively. Therefore, it can be concluded that hydrogen bonds are formed between NCS molecules and D−fructose molecules, and the D−fructose has a plastic effect on the NCS. Particularly, the absorption peak frequency is the smallest when the mass ratio of NCS to fructose is 10:2. At this mass ratio, the number of hydrogen bonds is the largest, and the hydrogen bonds are the most stable. When the fructose content continues to increase, hydrogen bonds tend to form between fructose molecules. Excessive accumulation of fructose in the starch matrix affects its plasticization effect. The D−fructose has a high molecular weight, and the number of –OH that can form hydrogen bonds with the starch is large. Therefore, D−fructose is less likely to precipitate after plasticization and can improve the stability of the starch matrix [24].

It is worth noting that both GLY and D−fructose have certain shortcomings when using each of them alone. GLY has a small molecular weight, and it can effectively penetrate starch granules. However, the GLY is also prone to precipitate, and the retrogradation problem of GLY−plasticized starch cannot be ignored [23]. The molecules of D−fructose can stabilize the starch matrix by restricting the fluidity and flexibility of starch chains and inhibiting recrystallization and retrogradation phenomena [25], but their molecular weight is too large to enter adjacent starch molecule chains. Therefore, combining both of them to produce a composite plasticizer is necessary.

To explore the synergistic plasticization ability of the composite plasticizer on NCS, NCS with different composite plasticizers are prepared, as shown in Figure 7c. When the mass ratio between NCS, GLY, and D−fructose changes from 10:1:1 to 10:2:1, the –OH stretching vibration peaks within the wavenumbers of 3645 cm^−1^~3300 cm^−1^ exhibit a trend of decreasing frequency, indicating that the plasticization effect of the composite plasticizer is better than that of single plasticizers. When the mass ratio between NCS, GLY, and D−fructose is 10:2:1, the vibration frequency of O–H is 3392.7 cm^−1^, and the absorption peaks show a clear trend of broadening. Therefore, the synergistic plasticization effect at this mass ratio is the best, and the greatest impact is achieved for the hydrogen bond structure of starch.

#### 3.2.2. Effects of Plasticizer on Crystallization Type

Figure 8a displays XRD images of TPS obtained using different proportions of NCS and GLY as plasticizers. NCS exhibits a combination of sharp crystal diffraction peaks and amorphous diffuse wide peaks, with its crystallinity primarily derived from amylopectin. The crystalline structures of NCS can be classified into A, B, and C types based on XRD analysis. Among them, the A−type crystal is highly ordered and belongs to the monoclinic crystal system, while the B−type crystal exhibits secondary order and belongs to the orthorhombic system. A C−type crystal represents a mixture of type A and type B, characterized by an unstable crystal structure [39]. NCS contains 72% amylopectin, which forms a stable double helix structure through hydrogen bonding and belongs to an A−type crystal. The main crystalline peaks are observed around 2θ = 15°, 17°, 18°, and 23°. When GLY is added as a plasticizer, it disrupts the original crystalline structure within the NCS, affecting the crystalline region. The original crystalline peaks (15°, 17°, 18°, and 23°) become weaker, and the dispersed broad peaks become dominant. The dispersed broad peaks represent the residual natural crystalline after the plasticization process of the NCS, and they indicate that the preparation of TPS is insufficient to completely disintegrate the starch granules [40]. Notably, the diffraction peak near 2θ = 19.8° represents a typical V_H_−type crystalline structure induced during the plasticization process [41]. This is attributed to the lubrication effect of small GLY molecules that penetrate and disrupt the starch granules. Under specific temperature and external force conditions, GLY infiltrates the interior of the starch, breaking the original hydrogen bonding and reducing the crystalline region. Over time, the branched starch molecules recombine to form a V_H_−type crystalline structure.

Figure 8b presents XRD images of TPS obtained using D−fructose at different proportions as a plasticizer. The images indicate that the crystalline structure of NCS modified by D−fructose is partially disrupted. The XRD pattern displayed sharp characteristic peaks near 17°, which can be attributed to the rearrangement of the double helix structure after NCS molecule dehydration. As the content of D−fructose increases, the intensity of the crystallization peak gradually weakens. When the starch−to−fructose mass ratio reaches 10:2, the diffraction peak becomes the weakest, and the peak at 15° completely disappears, indicating the strongest detrimental effect on the starch’s crystalline structure. This suggests that within a certain range of D−fructose content, it can form new hydrogen bonds with starch molecules, weaken the inherent hydrogen bonding of starch, disrupt the crystallization zone, and increase the number of free branched chains. Additionally, a new V_H_−type crystal peak appears near 20°, further confirming the disruption of the original starch crystal structure. It is important to note that D−fructose possesses more hydroxyl groups, larger spatial gaps with starch molecules, and increased cross−linking, enabling the formation of more stable hydrogen bonds with starch molecules. Furthermore, the C=O double bonds (Figure 1b) exhibit stronger electronegativity, resulting in denser electron clouds around oxygen atoms and facilitating the formation of new hydrogen bonds with starch molecules.

To enhance the plasticization effect, a combination of plasticizers, namely GLY and D−fructose, is used, as they individually have limited ability to break down starch granules. Figure 8c displays the XRD pattern of plasticized starch prepared using a composite plasticizer. When the mass ratio between NCS, GLY, and D−fructose is 10:2:1, no significant diffraction peak is observed. This indicates that the plasticization effect is optimal at this mass ratio, as it effectively disrupts the crystalline structure of starch.

#### 3.2.3. Effects of Plasticizer on Micromorphology

During the plasticization process of NCS, the crystal structure undergoes destruction, resulting in changes to the starch granules and their microscopic morphology. To further characterize the impact of plasticization modification on starch granules, it is necessary to analyze the changes in their microstructure. Figure 9 presents SEM images of three types of plasticized starch. Figure 9a illustrates the microstructure of NCS, which exhibits regular hexagonal−shaped granules with a smooth surface. The distribution of angularity is not prominent, and the overall shape is nearly spherical. Figure 9b displays the microstructure of GLY−plasticized starch, where the starch granules appear angular, elongated, and diamond−shaped, with a rough surface. Figure 9c showcases the microstructure of D−fructose−plasticized starch, revealing the clumping of starch granules and a small amount of granule detachment. Figure 9d presents the microstructure of GLY and D−fructose composite−plasticized starch, which demonstrates significant damage to the starch granules, characterized by evident holes, a rough surface, and raised small granules. In summary, the microstructure of NCS experiences noticeable damage when subjected to different plasticizers, with the most significant damage observed when composite plasticizers are employed.

### 3.3. Molecular Linking Model of Starch−Based Composites

To examine the cross−linking involving NCS, plasticizers, and plant fibers, a model illustrating hydrogen−bonding interactions of the physical network type is constructed, as depicted in Figure 10, rather than adopting covalent bond formations. Sisal fiber is subjected to alkalization treatment to dissolve the hemicellulose and pectin present in the fiber. During the alkalization process, sisal fiber reacts with NaOH, resulting in the formation of alkaline cellulose, as shown in Equation (4) [42]. Alkaline cellulose further ionizes to generate negatively charged alkaline cellulose anions, as depicted in Equation (5) [43]. When the –OH group on the plasticizer is in proximity to the O− group on the carboxymethyl of the fiber molecule, their strong magnetic fields attract each other, forming a new hydrogen bond. Furthermore, the free –OH group on the cellulose molecule also forms hydrogen bonds with the –OH group on the plasticizer. In summary, a stable structure of starch−plasticizer−fiber, with the plasticizers (GLY, D−fructose) serving as bridges, is formed during the preparation of the mixed slurry.
[C_6_H_7_O_2_(OH)_3_]n + nNaOH → [C_6_H_7_O_2_(OH)_2_ONa]n + nH_2_O(4)
[C_6_H_7_O_2_(OH)_2_ONa]n → [C_6_H_7_O_2_(OH)_2_O^−^]n + nNa^+^
(5)

### 3.4. Parameter Optimization of the Starch Composites

As an RSM, Box−Behnken design (BBD) is known for its high efficiency, processing speed, and simplicity and is widely utilized in industrial research. Table 3 presents the results obtained from the actual test based on the BBD. The results are used to fit a mathematical model to obtain a response surface, which is then utilized to determine the optimal process parameters.

The quadratic polynomial regression model (code) of the forming temperature (A), forming time (B), AC content (C), tensile strength (Y_1_), and rebound rate (Y_2_) is established by second−order polynomial fitting of experimental data. Y_1_ and Y_2_ can be expressed as follows:Y_1_ = 4.02 − 0.46A + 0.088B − 0.36C − 0.48AB + 0.15AC + 0.26BC − 0.44A^2^ − 0.49B^2^ − 0.60C^2^(6)
Y_2_ = 0.93 − 0.085A − 0.039B − 0.041C − 0.077AB − 0.031AC − 0.029BC − 0.081A^2^ − 0.019B^2^ − 0.059C^2^
(7)

An ANOVA was employed to assess the adequacy and significance of the second−order polynomial regression model. The results of variances for tensile strength and rebound rate are presented in Table 4 and Table 5, respectively. In these tables, the *p*−value and LOF are utilized to evaluate the validity of the regression model.

The *p*−value of the model serves to determine the influence degree of the parameters on the response. A lower *p*−value indicates a more significant influence of the parameter on the model, typically less than 0.05 [33]. In this investigation, the *p*−values of the regression model are 0.0049 and 0.0027, respectively, indicating high significance of the model.

LOF compares the disparity between the fitted surface and the actual value with the pure error to assess the capability of the model in representing the actual response surface. Ideally, the LOF should exceed 0.05 [44]. In this investigation, the *p*−values of the LOF in the regression model are 0.105 and 0.0632, respectively, indicating their lack of significance. Consequently, the model can adequately fit the experimental results.

Adept Precision (AP) represents the ratio of the difference between the maximum and minimum and predicted values to the average standard deviation of all predicted values. It serves as a measure of the signal−to−noise ratio, and an AP value higher than four is considered desirable [45]. In this investigation, the AP values are 7.773 and 10, respectively, indicating an adequate signal level.

Furthermore, the normality of the residuals is assessed through a normal probability graph of the residuals. When the number of models is adequate, the overall shape of the residual graph should resemble a straight line if the residuals follow a normal distribution [46]. Figure 11a,b displays the normal probability plots of the residuals for tensile strength and rebound rate, respectively. The graph depicting the relationship between residuals and the running order can be utilized to identify any correlation between residuals. If a correlation exists among residuals, the assumption of independence will be violated [47]. Figure 11c,d presents the effects of run number on residuals of tensile strength and rebound rate, respectively. The random distribution of points indicates that the independence assumption holds for the residuals in this model.

Figure 12 depicts the three−dimensional surface and contour plots of tensile strength and rebound rate. In Figure 12a,a′,d,d′, the AC content is kept constant at 1 g to investigate the effects of forming temperature and forming time on the mechanical properties of the samples. The results reveal that as the forming temperature and forming time increase, the tensile strength and rebound rate of the sample initially increase and then decrease. This behavior can be attributed to the fact that elevated forming temperatures enhance slurry fluidity, facilitate the decomposition of additives, and promote moisture volatilization within the material, thereby improving the forming efficiency. However, when the forming temperature becomes excessively high, the tensile strength of the sample remains low, and the rebound rate exhibits a declining trend with increasing forming time. This can be attributed to the detrimental effects of high temperatures on the quality of starch and sisal fiber, as well as the generation of uncontrollable gas. Prolonged exposure to high temperatures disrupts the internal structure of the material and adversely affects product quality. Similarly, if the forming time is too short, the slurry fails to mature adequately, resulting in incomplete material formation. Conversely, an excessively long forming time leads to starch and fiber charring, which negatively impacts the mechanical properties of the samples.

In Figure 12b,b′,e,e′, the impacts of forming temperature and AC content on the mechanical properties of the sample are examined with a fixed forming time of 5 min. The results indicate that both the tensile strength and rebound rate of the sample decrease as the forming temperature increases, with the minimum values obtained at the highest temperature. Regarding the influence of AC content, the tensile strength and rebound rate initially increase and then decrease with increasing AC content. AC, as the primary power source for the formation of the internal structure of composite materials, significantly affects product quality. Within a reasonable range, increasing the AC content allows for the attainment of a uniform and dense cell structure, which is favorable for foam formation. However, excessive or insufficient AC content hinders cell formation. When AC undergoes heat−induced decomposition, the resulting gas gradually dissolves in the melt/gas system. If the solubility of the system is exceeded, foaming occurs. During the initial stage of foaming, bubbles of different sizes are generated within the material, with smaller bubbles exerting higher internal pressure. Subsequently, the bubbles enter the growth stage, where they gather and merge upon contact, gradually forming stable voids. Insufficient AC content impedes foaming and hampers the formation and aggregation of bubbles. Conversely, excessive AC content leads to an excessive amount of gas, making bubble control challenging and increasing the likelihood of bubble rupture, thereby compromising the internal structure and adversely affecting the mechanical properties.

In Figure 12c,c′,f,f′, the impacts of forming time and AC content on the mechanical properties of the sample are examined at a fixed forming temperature of 205 °C. The results indicate that as the forming time and AC content increase, the tensile strength and rebound rate of the sample initially increase and then decrease, consistent with the findings of the T−t and T−AC experiments.

The optimization process is conducted using the Design Expert software. The optimal process parameters are as follows: the forming temperature (A) is 198.13 °C, the forming time (B) is 5.39 min, and the AC content (C) is 0.84 g. Based on the fitted second−order polynomial mathematical model, the predicted tensile strength is 4.28 MPa, and the rebound rate is 96%. Considering equipment limitations during actual processing, the specific process parameters selected for material preparation are as follows: forming temperature (A) is 198 °C, forming time (B) is 5.4 min, and AC content (C) is 0.84 g. The mechanical properties are then tested. The test results show that the actual process parameters result in a tensile strength of 4.22 MPa and a rebound rate of 95.7%, which are similar to the predicted values generated by the software and mathematical model. This demonstrates the reliability of the RSM for optimizing the forming process of starch−based packaging materials. The process parameters of group 3 in Table 3 are similar to those after optimization, so they are used as the control group without parameter optimization. In comparison with the non−optimized values, the tensile strength of the composite prepared with the optimized process parameters increases by 12.2%, and the rebound rate increases by 8.1%. Additionally, using the optimized process parameters, a starch−based biomass cell phone inner packaging box is prepared, as shown in Figure 13.

## 4. Conclusions

In this study, processable TPS is prepared by adding plasticizers. It has been observed that in the presence of plasticizers, the hydrogen bonds of starch are broken, leading to a reduction in crystalline peaks and disruption in the granule morphology. The use of composite plasticizers resulted in better effects. Subsequently, the preparation of starch−based packaging materials is carried out in a thermal cavity forming machine, and the optimization of the forming process parameters, including forming temperature, forming time, and AC content, is studied using RSM. It is observed that both the tensile strength and rebound rate exhibit an increasing trend followed by a decreasing trend with the increase of the parameters. The optimal mechanical properties are achieved at the process parameters of 198 °C forming temperature, 5.4 min forming time, and 0.84 g AC content. Compared to the non−optimized mechanical properties in the experimental design stage, the tensile strength increases by 12.2%, and the rebound rate increases by 8.1%.

## Figures and Tables

**Figure 1 materials-16-05953-f001:**
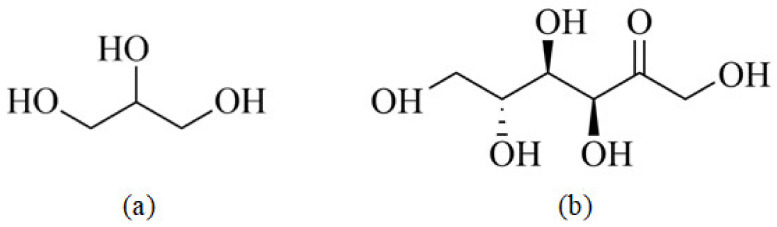
Molecular structures of plasticizers: (**a**) GLY and (**b**) D−fructose.

**Figure 2 materials-16-05953-f002:**
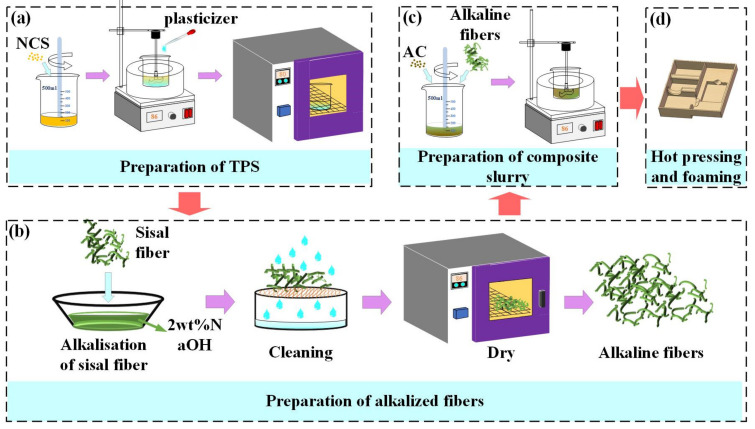
The preparation process of the starch−based packaging materials: (**a**) preparation of the TPS, (**b**) alkalization treatment of the sisal fibers, (**c**) preparation of the composite slurry, and (**d**) hot pressing and foaming.

**Figure 3 materials-16-05953-f003:**
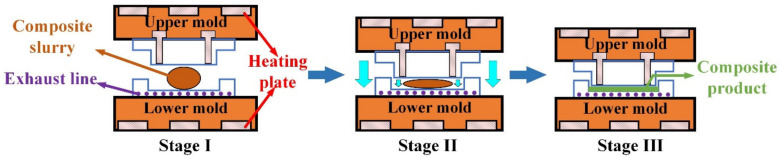
The experimental process of the thermal cavity−forming experimental device.

**Figure 4 materials-16-05953-f004:**
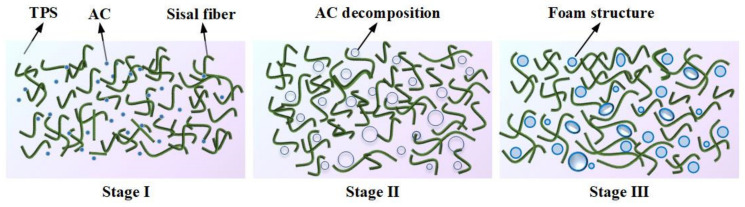
A diagram showing the three major stages of the foaming process.

**Figure 5 materials-16-05953-f005:**
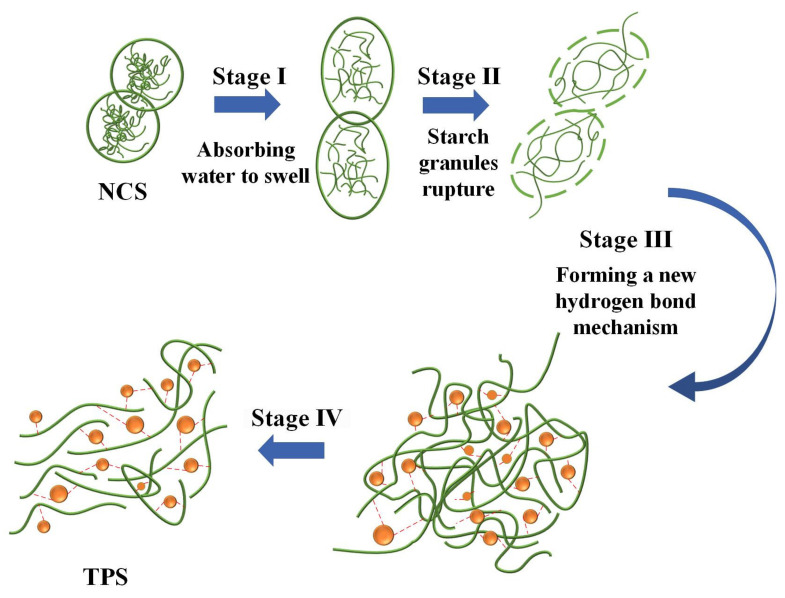
The four stages of the starch plasticization process.

**Figure 6 materials-16-05953-f006:**
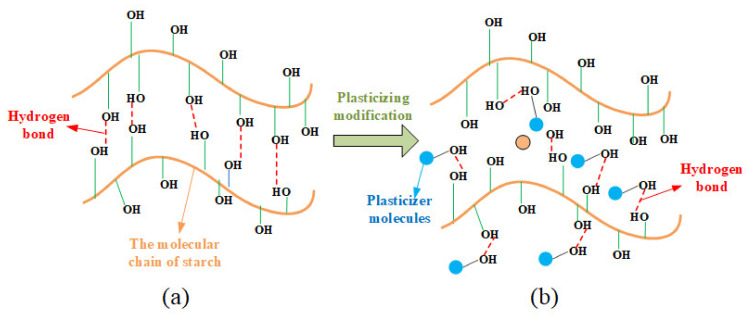
Plasticizer’s effects on disrupting the hydrogen bonding in starch: (**a**) Before adding the plasticizer, and (**b**) after adding the plasticizer.

**Figure 7 materials-16-05953-f007:**
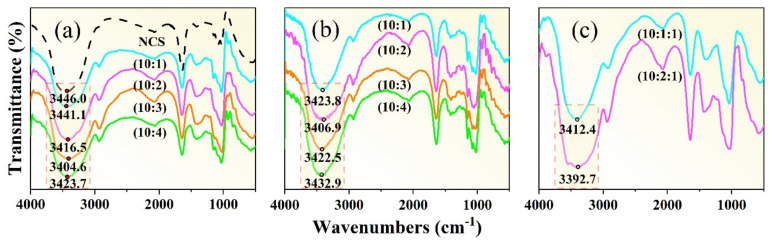
Infrared spectra: (**a**) Glycerol (GLY) is the plasticizer, (**b**) D−fructose (Fru) is the plasticizer, and (**c**) GLY and Fru are the composite plasticizer. The black line in figure (**a**) represents natural corn starch (NCS), and the blue line represents NCS:GLY = 10:1; the purple line represents NCS:GLY = 10:2; the orange line represents NCS:GLY = 10:3; the green line represents NCS:GLY = 10:4. Figures (**b**,**c**) are similar to figure (**a**).

**Figure 8 materials-16-05953-f008:**
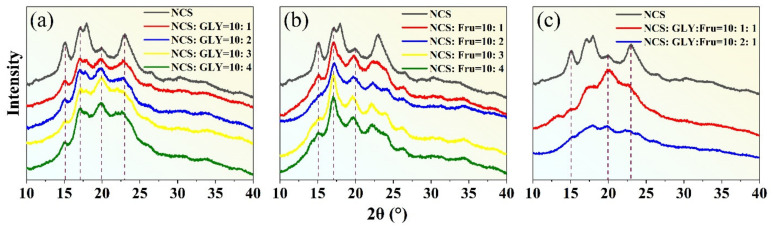
Effects of plasticizer content on XRD results when (**a**) glycerol (GLY) is the plasticizer, (**b**) D−fructose (Fru) is the plasticizer, and (**c**) GLY and Fru are the composite plasticizer. The black line in figure (**a**) represents natural corn starch (NCS), and the red line represents NCS:GLY = 10:1; The blue line represents NCS:GLY = 10:2; The yellow line represents NCS:GLY = 10:3; The green line represents NCS:GLY = 10:4. Figure (**b**,**c**) are similar to Figure (**a**).

**Figure 9 materials-16-05953-f009:**
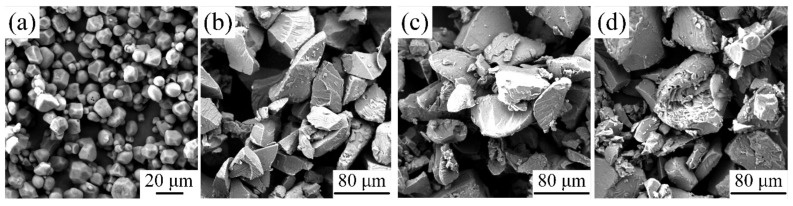
SEM images of the (**a**) natural corn starch (NCS), (**b**) NCS with glycerol (GLY) as the plasticizer, (**c**) NCS with D−fructose as the plasticizer, and (**d**) NCS with GLY and D−fructose as the composite plasticizer.

**Figure 10 materials-16-05953-f010:**
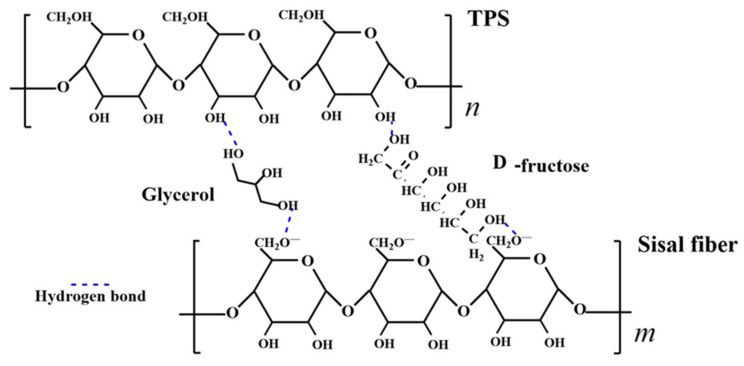
A molecular model of hydrogen bond interactions between TPS, plasticizers, and sisal fibers.

**Figure 11 materials-16-05953-f011:**
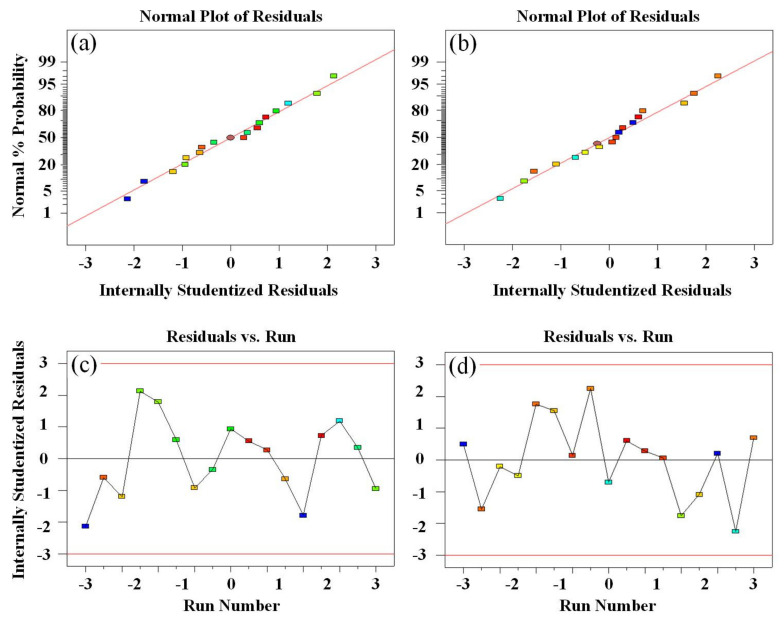
Tensile strength and rebound rate of (**a**,**b**) residual normal probability graph, and (**c**,**d**) diagram of the relationship between residuals and running sequence.

**Figure 12 materials-16-05953-f012:**
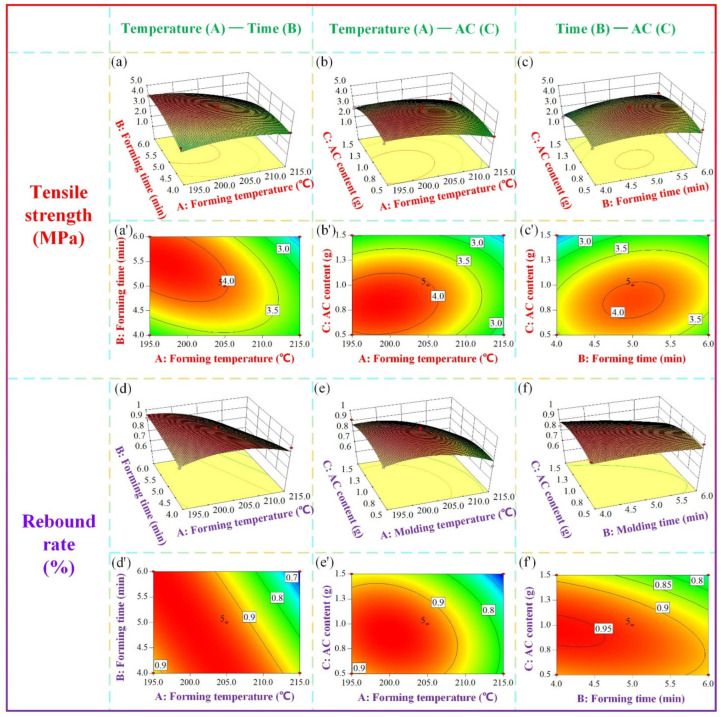
Response surface diagrams of (**a**–**c**) three−dimensional surface of tensile strength, (**a**′–**c**′) contour line of tensile strength, (**d**–**f**) three−dimensional surface of the rebound rate, and (**d**′–**f**′) contours of the rebound rate.

**Figure 13 materials-16-05953-f013:**
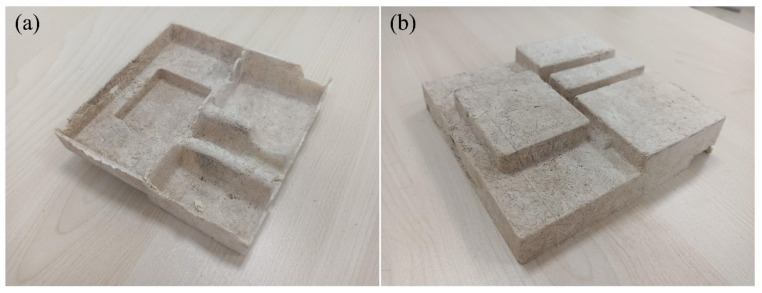
A cell phone packaging box made by starch−based biomass composite: (**a**) top view, and (**b**) bottom view.

**Table 1 materials-16-05953-t001:** Mass ratio of natural corn starch (NCS) to different plasticizers.

NCS:Glycerol	NCS:D−Fructose	NCS:Glycerol:D−Fructose
10:1	10:1	10:1:1
10:2	10:2	10:2:1
10:3	10:3	
10:4	10:4	

**Table 2 materials-16-05953-t002:** Different levels of process parameters.

Level	Forming Temperature (°C)	Forming Time (min)	AC Content (g)
−1	195 °C	4	0.5
0	205 °C	5	1
1	215 °C	6	1.5

**Table 3 materials-16-05953-t003:** Experimental design and results of the response surface.

Experiment Number	Forming Temperature (°C)	Forming Time (min)	AC Content (g)	Tensile Strength (MPa)	Rebound Rate (%)
1	215	6.00	1.00	1.90 ± 0.14	64.4 ± 1.3
2	195	6.00	1.00	4.02 ± 0.18	92.5 ± 2.1
3	195	5.00	0.50	3.76 ± 0.05	88.5 ± 0.9
4	195	4.00	1.00	3.33 ± 0.13	87.2 ± 1.1
5	205	6.00	0.50	3.4 ± 0.09	92.4 ± 1.2
6	215	4.00	1.00	3.11 ± 0.06	89.7 ± 0.4
7	205	5.00	1.00	3.76 ± 0.05	94.0 ± 0.6
8	195	5.00	1.50	2.88 ± 0.08	91.6 ± 0.6
9	205	6.00	1.50	3.06 ± 0.3	73.3 ± 2.5
10	205	5.00	1.00	4.18 ± 0.08	95.7 ± 0.5
11	205	5.00	1.00	4.10 ± 0.07	94.5 ± 0.4
12	205	5.00	1.00	3.84 ± 0.09	93.7 ± 0.7
13	205	4.00	1.50	1.93 ± 0.03	84.8 ± 0.2
14	205	5.00	1.00	4.23 ± 0.4	89.4 ± 2.7
15	215	5.00	1.50	2.49 ± 0.07	64.2 ± 0.4
16	215	5.00	0.50	2.79 ± 0.20	73.3 ± 1.7
17	205	4.00	0.50	3.31 ± 0.08	92.1 ± 0.6

**Table 4 materials-16-05953-t004:** Variance analysis of the second−order polynomial model for the tensile strength of samples.

Coefficient Source	Sum of Squares	Degree of Freedom	Variance	*p*−Value
model	7.84	9	0.87	0.0049
A− Forming temperature	1.71	1	1.71	0.0045
B− Forming time	0.061	1	0.061	0.4629
C−AC content	1.05	1	1.05	0.0147
AB	0.90	1	0.90	0.0205
AC	0.084	1	0.084	0.3931
BC	0.27	1	0.27	0.1468
A^2^	0.81	1	0.81	0.0257
B^2^	1.03	1	1.03	0.0155
C^2^	1.53	1	1.53	0.0060
Residual error	0.71	7	0.10	0.1050
LOF	0.53	3	0.18	
Error term	0.18	4	0.044	
Total variation	8.55	16		

**Table 5 materials-16-05953-t005:** Variance analysis of the second−order polynomial model for the rebound rate of samples.

Coefficient Source	Sum of Squares	Degree of Freedom	Variance	*p*−Value
model	0.16	9	0.018	0.0027
A− Forming temperature	0.058	1	0.058	0.0007
B− Forming time	0.012	1	0.012	0.0327
C−AC content	0.013	1	0.013	0.0282
AB	0.024	1	0.024	0.0078
AC	0.0037	1	0.0037	0.1854
BC	0.0035	1	0.0035	0.1984
A^2^	0.028	1	0.028	0.0051
B^2^	0.0015	1	0.0015	0.3839
C^2^	0.015	1	0.015	0.0220
Residual error	0.012	7	0.0017	
LOF	0.0098	3	0.0033	0.0632
Error term	0.0022	4	0.0057	
Total variation	0.17	16		

## Data Availability

Data will be made available on request.

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
