# Peer review of "Enhancing Starch−Based Packaging Materials: Optimization of Plasticizers and Process Parameters"

_materials, 2023, doi:10.3390/ma16175953_

Round 1

Reviewer 1 Report

The paper deals with the production of starch-based packing materials, namely on the base of thermoplastic starch and glycerol and D-fructose used as plasticizers. For preparing of curing composite materials the sisal fibers were used.

It is necessary to note that in the literature there is a lot of works dedicated to creation and investigation of numerous composite materials on the base of starch with different plasticizers. So, the obtained by authors results concerning of the influence of plasticizers on the structure and mechanical properties of starch are well known.

 In common case for preparing of reinforced materials on the base of starch the different synthetic fibers are used. The peculiarity of presented work is the applying of nature origin sisal fibers as curing agent, so the obtained compositions is fully biodegradable that made it perspective material for further application, although the increasing of the tensile strength on 12,2 % is not very significant.

But it is obviously that in the case of use of synthetic fibers even such reкsult could not be achieved due to the incompatibility of polysaccharide starch with such synthetic fibers as ethylene, polypropylene etc.

Nevertheless the obtained materials obviously may be used for construction of some biodegradable items.

On my opinion due to the enough good scientific level of presented work the paper may be published in journal Materials.   

Reviewer 2 Report

121   change "plastic agent" to "plasticizing agent"

136   "the starch is completely dissolved"  The starch do not dissolves in water below 70-80 °C ¡¡¡¡

158 "as the temperature increases, the viscosity of TPS gradually rises."

What are the authors based on to affirm this?

194 "The formula for calculating the rebound rate is:"

How the rebound rate equation is obtained ? Include its deduction.

386-389 "When the mass ratio between NCS, GLY, and D-fructose is 10:2:1, no significant diffraction peak is observed. This indicates that the plasticization effect is optimal at this mass ratio, as it effectively disrupts the crystalline structure of starch and inhibits starch retrogradation."

Does the authors have experimental information that justifies that inhibits the starch retrogradation?  What are the authors based on to affirm this?

399 The SEM micrographs (fig 9) correspond to granules of pure starch (NCS), and mixed with glycerol, fructose and the combination of both? The granule sizes shown are very different. What is its size distribution?

It is necessary to include SEM micrographs of the TPS films with the different plasticizers where the starch granules that were not plasticized can be observed.

411  "3.3. Molecular linking model of starch-based composites"

In this section, it must be clearly established that the hydrogen bond-type interactions established between plasticizers and starch and possibly with agave fibers would form a physical network and not covalent bonds, which is how the term crosslinking is generally used.

Reviewer 3 Report

In this study, a new starch-based packaging material is developed by adding plasticizers to it. The process has been optimized by RSM. This approach effectively reduced crystallinity and improved tensile strength and rebound rate.

The research is original but the novelty or the gap in the previous studies has not been discussed well. My first comment for the improvement of this paper is about this.

This study is using a well-known statistical approach for the optimization which can be considered as the main addition of this study into the area.

The methodology is well presented. It will be better to add the standards used for the characterizations.

The conclusions are consistent with the evidence and arguments presented and they address the main question posed. The references are appropriate and suggested some addition regarding the discussion about the statistical analyses.

1-    Indicate the novelty of this study clearly and compare with the similar studies.

2-    In statistical analysis, more discussion using the recent publications is recommended. Following are some recommended articles:

a.     https://doi.org/10.1111/cote.12651

b.     https://doi.org/10.1007/s11694-022-01632-7

c.     https://doi.org/10.22201/icat.24486736e.2022.20.4.1239

Minor revision will improve it.

Reviewer 4 Report

In this manuscript, the authors investigated composite packaging material based on thermoplastic starch using glycerol, D- fructose, and a mixture of them as the plasticizer optimized by the response surface method. The manuscript presents interesting results but needs substantial improvement before it can be considered for publication in the Materials, as I can list some of them below:

1. The abstract contains a quite long unnecessary introduction before some parts of the main results. A brief introduction would be enough and for example, the preparation method and the kind of filler should be mentioned. Therefore, the authors must rewrite this part and add more important results in this section. Moreover, the authors should add more important results in this section as well. The abbreviations must be defined at their first mention in the abstract itself (AC).  

2. Please, reconsider the first keyword "Plasticizer plasticizing".

3. Introduction needs to be improved and restructured to show better the work's originality and importance. Moreover, it seems that the novelty of the work with respect to the literature (recent works), and the state of the art are not well highlighted in the paper. For this purpose, you can use the following references: Carbohydrate Polymers 269 (2021) 118250, Polymers 2021, 13, 3819, Polymer International 2020; 69: 317–327, Journal of Polymer Research (2022) 29:257, and Materials 2023, 16(3), 900.  Also, a clear statement on what was intended to be investigated (expectations, aims, hypotheses) would be helpful (line 106 is quite vague).

4. I highly recommend introducing and having a short explanation about the most common plasticizers used in preparing starch-based biodegradable packaging material in the introduction section.

5. It is well known that moisture content and relative humidity environments play a significant role in the gelatinization and retrogradation (crystallization) of starch. Thus, what conditions were used for the prepared samples, especially for the mechanical properties measurements? It should be mentioned in the materials and methods section. Moreover, did the authors consider investigating the properties of the new packaging material after storage?

6. Standard deviations also should be provided given that it is an important feature of the work

7. Let the table and figure legends be more informative (especially abbreviations and sample codes) to avoid the readers getting back to the methodology section to understand what we can see in the results.

8. There are many grammatical errors. Please have a thorough proofread of the entire manuscript for proper English usage. 

There are many grammatical errors. Please have a thorough proofread of the entire manuscript for proper English usage.

Round 2

Reviewer 4 Report

The authors have taken my comments seriously and made proper revisions accordingly. The revised manuscript is now suitable in terms of its technical quality and originality for publication in Materials